# Low-Noise Amplifier with Bypass for 5G New Radio Frequency n77 Band and n79 Band in Radio Frequency Silicon on Insulator Complementary Metal–Oxide Semiconductor Technology

**DOI:** 10.3390/s24020568

**Published:** 2024-01-16

**Authors:** Min-Su Kim, Sang-Sun Yoo

**Affiliations:** 1Department of Information and Electronic Engineering, Mokpo National University, Muan 58554, Republic of Korea; msmy970@mnu.ac.kr; 2Department of Electrical and Computer Engineering, Sungkyunkwan University, Suwon 16419, Republic of Korea

**Keywords:** fifth generation (5G), new radio (NR), low-noise amplifier (LNA), RC feedback, series inductive-peaking, bypass mode, wideband

## Abstract

This paper presents the design of a low-noise amplifier (LNA) with a bypass mode for the n77/79 bands in 5G New Radio (NR). The proposed LNA integrates internal matching networks for both input and output, combining two LNAs for the n77 and n79 bands into a single chip. Additionally, a bypass mode is integrated to accommodate the flexible operation of the receiving system in response to varying input signal levels. For each frequency band, we designed a low-noise amplifier for the n77 band to expand the bandwidth to 900 MHz (3.3 GHz to 4.2 GHz) using resistive–capacitance (RC) feedback and series inductive-peaking techniques. For the n79 band, only the RC feedback technique was employed to optimize the performance of the LNA for its 600 MHz bandwidth (4.4 GHz to 5.0 GHz). Because wideband techniques can lead to a trade-off between gain and noise, causing potential degradation in noise performance, appropriate bandwidth design becomes crucial. The designed n77 band low-noise amplifier achieved a simulated gain of 22.6 dB and a noise figure of 1.7 dB. Similarly, the n79 band exhibited a gain of 21.1 dB and a noise figure of 1.5 dB with a current consumption of 10 mA at a 1.2 supply voltage. The bypass mode was designed with S_21_ of −3.7 dB and −5.0 dB for n77 and n79, respectively.

## 1. Introduction

Currently, the 3rd Generation Partnership Project (3GPP) is preparing for 5G to address the challenges of saturated communication systems, aiming for high system capacity, wide coverage, and efficient data transmission [1]. For the frequency allocation in New Radio (NR), both licensed and unlicensed spectra from 1 GHz to 52.6 GHz are being considered. Particularly, millimeter-wave frequencies are attracting attention for their advantages in providing increased bandwidth, high data capacity, and rapid transmission [2]. However, the use of millimeter-wave frequencies poses challenges such as signal loss and the necessity for complex infrastructure setups using multi-antenna systems. Furthermore, designing additional stability techniques and implementing multi-stage amplifiers are essential to achieve both high gain and stability. However, this results in an overall increase in the complexity of the system [3]. In contrast, 5G systems in the sub-6 GHz band can utilize system infrastructure using the existing 4G LTE band, and as a technology already implemented in various countries, it proves advantageous in meeting the rapidly growing demands for high data rates. In particular, the United States has adopted 5G, utilizing NR bands such as 2.5–3.98 GHz and 4.49–4.99 GHz [4]. Furthermore, several other countries are also implementing 5G in the n77/79 bands, underscoring the growing emphasis on the need for research to optimize system performance within this frequency range.

Cascode low-noise amplifiers are commonly applied in applications requiring high gain and low noise characteristics within a narrow bandwidth [5]. However, for Cascode low-noise amplifiers intended for 5G NR, where bandwidth limitations exist, techniques for bandwidth extension become essential to achieve the desired bandwidth. Various techniques, such as distributed, coupled/resistive feedback, series peaking, double L-type matching network, and gm-boosting, are employed for this purpose [6,7,8,9,10,11]. In the case of the representative RC feedback technique, the bandwidth of the low-noise amplifier is determined by the transistor’s Ft (transition frequency) and the feedback loop. However, relying solely on feedback for bandwidth extension can result in a significant degradation in noise performance, making it challenging to optimize for low noise. To address this issue, additional techniques are employed to minimize performance degradation. References [7,8,9] demonstrate excellent performance over a wide bandwidth but exhibit a high noise figure of 4.5 dB. Alternative methods, peaking techniques, involve additional components such as inductors or T-coils and serial/parallel connections for bandwidth extension. However, these methods also come with the potential drawback of performance degradation in noise figures due to the broad bandwidth. Excessive broadband expansion techniques can, thus, be potential factors in performance degradation. Therefore, for low-noise amplifiers designed for 5G applications requiring a bandwidth of 600 MHz, it becomes crucial to avoid techniques that are overly wideband, considering the trade-off relationship among performance factors.

This paper presents the design of a low-noise amplifier (LNA) for 5G NR. Additionally, the system configuration was simplified by integrating LNAs for two bands into one chip. The designed amplifier integrated both the internal input and output matching networks, eliminating the need for an external matching circuit. The fabrication process utilized RF silicon on insulator (SOI) technology, known for its superior RF performance compared to bulk-CMOS processes. For the most suitable bandwidth expansion technique for the n77 and n79 bands, we proposed performance optimization using resistive feedback and inductively peaking techniques. In Section 2, we discuss selective bandwidth extension methods considering the bandwidths of each band, and in Section 3, we explain the simulation results of the optimized low-noise amplifier and conclusion.

## 2. LNA Design Using RC Feedback and Inductively Peaking Technique

Figure 1 presents the circuit diagram and simplified equivalent circuit of the common source with resistive feedback (R_FB_). The low-noise amplifier in Figure 1a employs a structure with a degeneration inductor, enabling simultaneous matching for input and noise and making it widely utilized. In Figure 1b, a low-noise amplifier with resistance feedback is represented, simplified as a single-stage CS (common-source) amplifier. This configuration yields the following impedance and noise figure performance [8].
(1)$$\begin{array}{c} {\;Z}_{IN,wFB}=\frac{1}{{sC}_{g}}||\left[{sL}_{S}+\left(\frac{1}{{sC}_{gs}}||\;R_{f}\right)\right]=\frac{1}{{sC}_{g}}||\frac{s^{2}L_{g}C_{gs}R_{f}+{sL}_{g}+R_{f}}{{sC}_{gs}R_{f}+1}\\ =\frac{s^{2}L_{g}C_{gs}R_{f}+sL_{g}+R_{f}}{s^{3}L_{g}C_{gs}C_{g}R_{f}+s^{2}L_{g}C_{g}+sR_{f}\left(C_{g}+C_{gs}\right)+1} \end{array}$$
(2)$$NF\approx 1+\frac{4Rs}{R_{FB}}+\gamma +\gamma g_{m}R_{s}\;$$

In Equation (1), C_gs_ represents the intrinsic gate-source capacitance, L_s_ is the degeneration inductor, L_g_ is the input inductor for matching, and R_FB_ is composed for feedback. As depicted in the equation, the input impedance decreases with the resistance value of R_FB_, leading to an extended bandwidth of the low-noise amplifier. However, as indicated in Equation (2), an increase in the feedback resistor (R_FB_) also increases the noise figure [11]. Moreover, the bandwidth extension technique using inductive peaking has the effect of compensating for the parasitic capacitance (C_parasitic_) occurring between drain and source, as illustrated in the simplified equivalent circuit shown in Figure 1b, thereby increasing the F_T_ of the transistor. The series-peaking inductor introduced in this process can be represented as a simplified transimpedance, as shown in the equation below [10].
(3)$${\;Z}_{N}=\frac{1}{1+\frac{s}{\omega_{0}}+\left(\frac{1-k_{c}}{m}\right)\frac{s^{2}}{\omega_{0}^{2}}+\left(\frac{k_{c}(1-k_{c})}{m}\right)\frac{s^{3}}{\omega_{0}^{3}}}\;$$

The parameter K_c_ represents the ratio of parasitic capacitance (C_parasitic_) to the output capacitance (C_output_), indicating the proportion of all capacitances generated at the parasitic and output stages. The bandwidth, as discussed in reference [10], increases at specific frequencies depending on the ratio of parasitic capacitance and the inductor used for compensation. For this purpose, bandwidth expansion is proposed through a peaking technique using T-coil, offering the advantage of adaptive operation according to output. However, complexity increases with the use of the output T-coil transformer for a wide bandwidth, and overall noise increases due to the wide bandwidth. As illustrated in Figure 1b, the series inductively peaking technique, known as a method of expanding bandwidth without using a transformer structure, compensates for the Cds parasitic component of the transistor. It is a technology that can expand bandwidth through this method. At this point, the impedance for the output can be expressed as follows.
(4)$$Z_{OUT}=\frac{1}{{sC}_{parasitics}}||{sL}_{compensation}=\frac{{sL}_{Lcompensation}}{{s^{2}C}_{parasitics}L_{compensation}+1}$$

As shown in (4), the output impedance can be compensated using the L_compensation_, and the bandwidth can be expanded at the desired frequency. However, this expansion technique has the characteristic of making it challenging to secure matching conditions for a wide bandwidth, rendering it suitable as a technique for a moderate bandwidth.

To select an appropriate bandwidth extension technique for the low-noise amplifier (LNA) designed for 5G NR in Figure 2, simulations were conducted based on a Cascode LNA with a degeneration inductor, emphasizing high gain and isolation characteristics. To minimize performance variations, the values of L_g_, L_S_, and L_L_ for matching conditions were optimized according to bandwidth techniques, and all inductors were set to have a quality factor of 20. The simulations were performed at frequencies where S_21_ gain exhibited a 1 dB degradation point, and matching conditions achieved S_11_/S_22_ of −10 dB.

The n77 band spans from 3.3 GHz to 4.2 GHz, providing a wide bandwidth of 900 MHz. To achieve this, various techniques, including Cascode LNA with RC feedback and inductive peaking, were applied to extend the bandwidth. The core transistors utilized had W/L ratios of 2 u/90 n, while the Cascode transistors had W/L ratios of 2 u/180 n. The simulations were performed using ideal resistors, capacitors, and inductors at a reference current of 10 mA.

Table 1 summarizes the simulation results using RC feedback or inductor peaking for the n77 band. While the typical Cascode LNA1 exhibits high gain and low noise characteristics, it falls short of providing the required bandwidth for the expansive 900 MHz n77 band. However, LNAs employing RC feedback and inductor peaking techniques show simulated gains of 22.2 dB and 22.5 dB, with noise figures of 1.39 dB and 1.27 dB, respectively. Regarding bandwidth, LNA2 and LNA3 designs achieve a wide bandwidth exceeding 1.6 GHz. Notably, incorporating inductor peaking in the feedback yields an additional margin in bandwidth for S_11_ and S_22_. For RC feedback, designing with lower resistance values for S_11_ and S_22_ is possible, but this compromises NF performance. Therefore, for the n77 band, the optimal strategy involves harmoniously employing feedback with relatively higher resistance values and inductor peaking.

Table 2 summarizes the simulation results for the n79 band. The reason for achieving higher gain and NF characteristics compared to the n77 band is that the n77 band inherently has a higher gain than n79. Therefore, due to the need for more feedback, the noise characteristics degrade more significantly. For the n79 band, with a bandwidth of 600 MHz from 4.4 GHz to 5 GHz, it allows for relatively low feedback to be applied. As observed in the results, it was possible to satisfy the bandwidth requirements for 600 MHz even without applying inductor peaking, using only RC feedback. This implies that for low-noise amplifiers designed for the n77 band, both RC feedback and inductor peaking are essential, while for the n79 band, satisfactory performance in gain and NF can be achieved with RC feedback alone.

## 3. Design and Simulation

Figure 3 shows the circuit diagram of the designed LNA, incorporating selected bandwidth extension techniques for optimization. Additionally, for the flexible operation of the receiver system at high input power, a design with symmetric T-type switches in bypass mode is implemented. The designed low-noise amplifier includes protection circuits for electrostatic discharge (ESD) at both input and output and integrates both input and output matching networks internally. For the n77 band, designed to achieve broadband performance compared to n79, a multi-section input matching (L_p_ and L_g_) is employed, as illustrated in Figure 3a. The table below indicates the values of RC feedback and the inductive-peaking technique applied to each band, tailored for optimal performance in terms of noise figure (NF) and bandwidth. Table 3 summarizes the sizes of devices used for the simulation.

Figure 4 illustrates the simulated S_11_, S_21_, S_22_, and noise figure (NF), and simulation results were all based on a full-PEX (parasitic extraction). The LNA consumes 10 mA at 1.2 supply voltage. As shown in Figure 4a, both S_11_ and S_22_ meet the −10 dB specification in the desired bands for both LNAs. The S_21_ for n77 is 22.6 dB, while for n79, it is 21.1 dB. Figure 4b shows the simulated NF, with n77 designed at 1.71 dB and n79 at 1.52 dB. Additionally, Figure 4c shows the s-parameter results for bypass mode. When a high input signal flows into the LNA, the designed LNA can adjust the gain from −4.48 to −3.92 dB at the n79 band using the bypass mode, and the n77 band is designed to be −5.16 to −4.23 dB. At this time, NF showed results of 4.28 to 4.31 dB and 4.93 to 4.05 dB in the n77 and n79 bands. The values of the elements used in the design are shown in Table 3.

Figure 5 is designed at 1500 × 1100 um^2^ using 90 nm RF-SOI CMOS technology and includes a bump pad, input and output matching, and an ESD protection circuit. Additionally, the inductor used in the design was custom-designed to minimize the area using the EMX simulator. For performance comparison, we utilized the figure of merit (FOM) in Equation (5), incorporating area (mm^2^), power consumption (P_mW_), noise figure (NF), and gain of S_21_, as depicted in the equation below.
(5)$$FOM=\frac{S_{21,dB}}{\left(NF-1\right)\times P_{mW}\times Area}\;$$

Table 4 presents a comparison with the results of previous papers. The proposed low-noise amplifier integrates the n77 and n79 bands into one chip, with each LNA showing a power consumption of 12 mW, and the size of the integrated chip is 1.5 mm^2^. Based on the figure of merit (FOM) calculated in Table 4, the proposed LNA shows comparable performances to the other designs.

As summarized in Table 4, low-noise amplifiers designed to cover multiple bands often employ broadband techniques. However, the gain and noise performance in these designs may not be optimized for sub-6 GHz. In contrast, the proposed low-noise amplifier exhibits both bandwidth and optimal gain and noise performance specifically tailored for the n77 and n79 bands. Furthermore, it offers the advantage of integrating two low-noise amplifiers on a single chip.

## 4. Conclusions

This paper proposes a highly integrated design for low-noise amplifiers (LNAs), matching circuits for input and output, and electrostatic discharge (ESD) protection circuits specifically tailored for the n77 and n79 bands. The integration of circuits for input and output matching offers advantages such as cost savings and minimal performance degradation in response to external environmental changes, eliminating the need for additional circuits for external components. Additionally, a bypass mode is incorporated to bypass the gain stage at high input signals, preventing system saturation. For the low-noise amplifier targeting the n77 band, a bandwidth extension design utilizing RC feedback and a series inductive-peaking technique was employed to cover the 900 MHz range (3.3 GHz to 4.2 GHz). Similarly, for the n79 band with a 600 MHz bandwidth (4.4 GHz to 5.0 GHz), an RC feedback technique was applied to optimize the performance of the low-noise amplifier. Through the use of band extension techniques tailored to the desired bands, the design achieves optimal performance, as simulated.

## Figures and Tables

**Figure 1 sensors-24-00568-f001:**
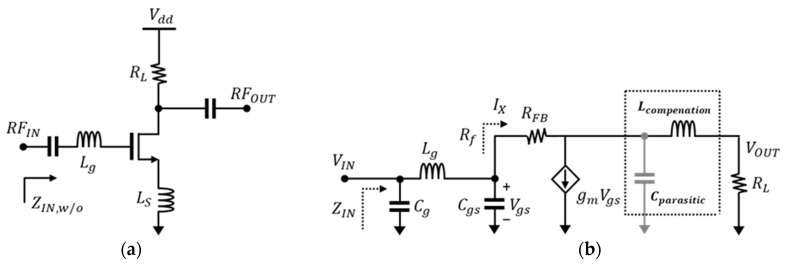
The general common-source LNA (**a**) without bandwidth expansion technique, and (**b**) equivalent circuit with resistive feedback and inductive-peaking technique.

**Figure 2 sensors-24-00568-f002:**
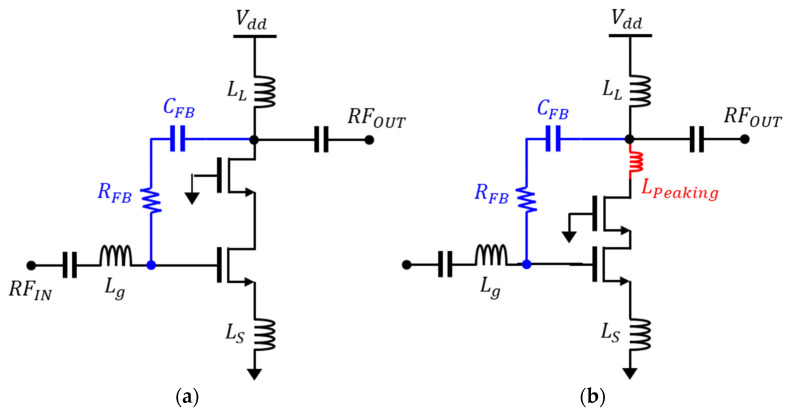
Low-noise amplifiers with applied bandwidth extension techniques for the n77 and n79 bands: (**a**) RC feedback and (**b**) RC feedback with inductor peaking.

**Figure 3 sensors-24-00568-f003:**
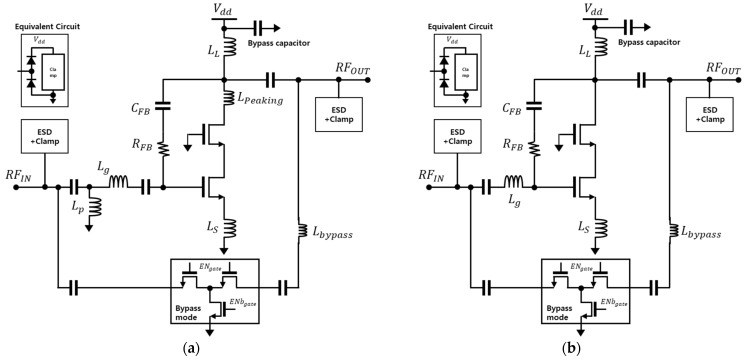
Low-noise amplifiers with bypass (**a**) for n77 band and (**b**) n79 band.

**Figure 4 sensors-24-00568-f004:**
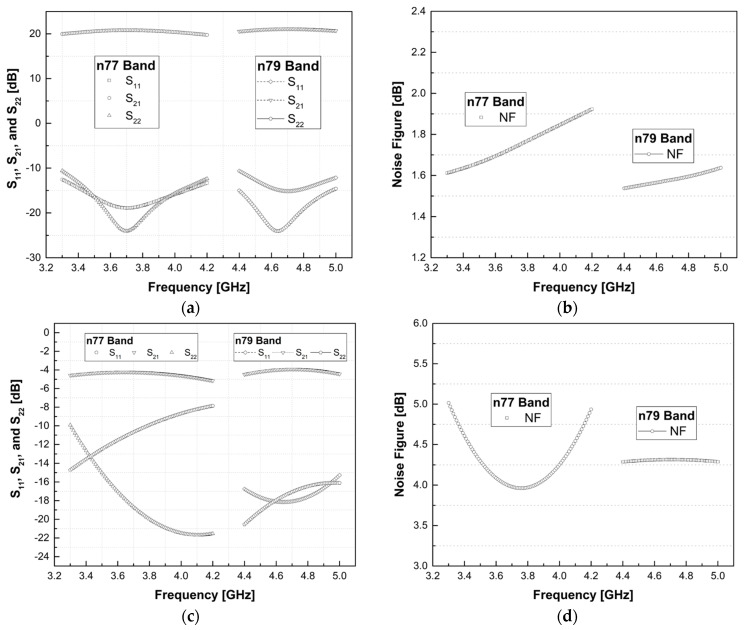
Simulation results for n77 and n79: (**a**) high gain mode of S_11_, S_21_, S_22_, (**b**) NF, and (**c**) bypass mode of S_11_, S_21_, S_22_, (**d**) NF.

**Figure 5 sensors-24-00568-f005:**
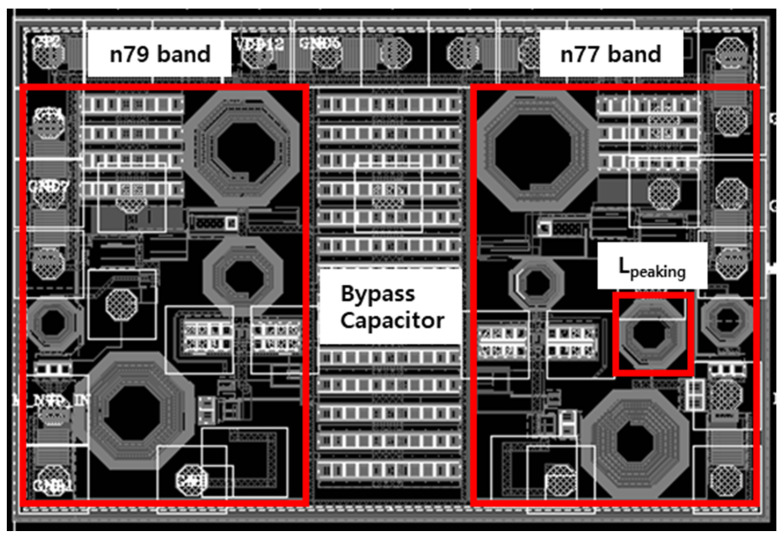
The layout of the designed LNA for n77/79 bands.

**Table 1 sensors-24-00568-t001:** Simulated results for n77 band.

Type	S_21_ (dB) _@3.75 GHz_	NF (dB) _@3.75 GHz_	BW_@Gain-1 dB_	BW_@S11/22 −10 dB_
LNA1 ^#^	23.9	0.95	600 MHz	500 MHz
LNA2 ^##^	22.2	1.39	1600 MHz	900 MHz
LNA3 ^###^	22.5	1.27	1600 MHz	1000 MHz

^#^ Cascode LNA without techniques, ^##^ RC feedback, and ^###^ RC feedback and inductive-peaking.

**Table 2 sensors-24-00568-t002:** Simulated results for n79 band.

Type	S_21_ (dB) _@4.7 GHz_	NF (dB) _@4.7 GHz_	BW_@Gain-1 dB_	BW_@S11/22 −10 dB_
LNA1 ^#^	23.8	0.84	600 MHz	400 MHz
LNA2 ^##^	24.3	1.02	1200 MHz	1200 MHz
LNA3 ^###^	24.3	1.00	1300 MHz	1400 MHz

^#^ Cascode LNA without techniques, ^##^ RC feedback, and ^###^ RC feedback and inductive peaking.

**Table 3 sensors-24-00568-t003:** Summary of device size and values.

Mode	Band	Core Size (um)	Cascode Size (um)	FB (KΩ/pF)	Ind.-Peaking (nH)
High gain mode	n77	W/L = 2 u/90 n	W/L = 2 u/180 n	R = 5.10/C = 0.5	0.8
	n79			R = 6.98/C = 0.5	-
Bypass mode	n77	W/L = 6 u/280 n		L_bypass_ = 0.65 nH	
	n79				

**Table 4 sensors-24-00568-t004:** Summary of performance comparison with previous papers.

Ref.	Technology	Frequency /BW (GHz)	S_11_/S_22_ (dB)	Gain (dB)	NF (dB)	Power (mW)	Area (mm^2^)	FOM
[4]	180 nm CMOS	3–8/5	<−10	16.4	2.9–4.66	3.9	0.62	3.5~1.8
[6]	180 nm CMOS	1.3–12.3/11	-	8.2	4.6–5.5	4.5	1	0.5~0.4
[9]	110 nm SOI CMOS	3–4.5 */1.5	−8/−7	15.2	1.0–1.56	16.1	0.18	~9.3
[12]	90 nm CMOS	3.1–10.6/7.5	<−10.6/−6	19.2–20.8	2.3–3.7	5.05	0.4941	5.9~3.08
[13] ^#^	90 nm CMOS	0–20/20	<−19.8	17.2	1.88	3.6	-	-
This work ^#^	90 nm RF-SOI	3.3–4.2/0.9 4.4–5.0/0.6	<−8	19.9–19.7 20.5–20.6	1.6–1.9 1.5–1.6	12 12	0.75 0.75	3.6~2.4 4.5~3.8

* Calculated from plot, ^#^ simulated result.

## Data Availability

Data are contained within the article.

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
