# Peer review of "Low-Noise Amplifier with Bypass for 5G New Radio Frequency n77 Band and n79 Band in Radio Frequency Silicon on Insulator Complementary Metal–Oxide Semiconductor Technology"

_sensors, 2024, doi:10.3390/s24020568_

Round 1

Reviewer 1 Report

Comments and Suggestions for Authors

1-Introduction section should be improved. Provide more context: The paper could benefit from providing more background information on 5G NR systems, the importance of low-noise amplifiers, and the current state of research in this area. Also, at least 5-6 new similar LNA should be investigated in this paper.

2-References should be updated and some brand-new related works should be added.

3-Comparison table should be improved and proposed design should be compares with LNAs with same technology (90nm)

4- Modify Table 4, and Provide BW in table. This table should be expanded.

5. Discuss limitations and future work: The paper should acknowledge any limitations of the proposed low-noise amplifier and suggest areas for future work and improvement.

6- The applied bandwidth extension techniques in this paper is not new. Provide the novelty of paper clearly.

7- In this design several inductors are used, which is not desirable. Provide explanations about draw backs.

8- The device is not fabricated. Can you provide explanations about validations of provided data.

Comments on the Quality of English Language

Minor editing of English language required

Author Response

Thank you for your good comments. Share the review report.

Reviewer 2 Report

Comments and Suggestions for Authors

The quality of the English is sufficient but could be improved. I highlighted some passages in the attached PDF. As to the content and readability of the paper, my comments follow.

1.

The logical flow of the Section 2 is rather "limp," so to say. The title suggests that an overview of bandwidth expansion techniques will be provided. However, the only one which is presented from a general standpoint is the inductive-peaking technique, as applied to a cascode stage. Then, the Authors simply switch to the cascode topology, without explaining how/justifying why the previous results should transfer to the latter. Even if the approach to the study of the cascode topology is through simulations (as it seems) the change of topic should be more clearly marked. At present, the reader is presented with a choice among possible techniques ("To select an appropriate bandwidth extension technique...") as if multiple possibilities had already been offered, which is not the case.

2.

Within the analysis of the common-source topology, the general architecture is presented in Fig. 1(a), whereas the small-signal circuit in Fig. 2(b) refers to the circuit with the technique applied: I strongly suggest either one of the following approaches:

- using two separate figures, so that the large-signal and small-signal circuit match;

- presenting the more particular topology, including the components for inductive-peaking, where these are made easily identifiable by means of dashed enclosures (similarly to what is already done in Fig. 2(b)).

3.

Tables 1 and 2 compare simulations, but details about the circuit (active and passive components) are missing: this information is partly proveded in Section 3 but should be anticipated to let the reader frame the results of the simulations. Also, the three circuits are referred to through numbers in square brackets, which is the same notation as for references: please differentiate.

4.

The comparison with the state of the art is not adequately discussed: a FOM is presented and the obtained performance is claimed to be "comparable." Please expand the comparison, addressing both the strong and weak points of the proposed design as compared to the literature. This is usually good practice but is even more important in the present case, where the FOM chosen by the Authors as significant does not seem particularly favorable to their design: two out of four designs (including the Authors') show a FOM which is approximately 2 to 8 times better.

5.

In any case, the given FOM is not merged into a discussion but simply "shown" before the text which uses it, as if it were a picture. The parameters used are not presented. In particular, I guess F is noise factor, i.e., according to typical convention, the quantity which, when expressed in dB, becomes the noise figure NF. If so, by the way, the Authors are using two different symbols for the same quantity: see eq. 2, where noise factor is denoted with NF as opposed to with F (the latter being recommended).

Comments on the Quality of English Language

The quality of the English is sufficient but could be improved. I highlighted some passages in the attached PDF.

Author Response

(The authors gave the same response as above.)

Round 2

Reviewer 1 Report

Comments and Suggestions for Authors

The authors have addressed most of my concerns.

Comments on the Quality of English Language

Minor editing of English language required

Author Response

First of all, thanks for the review. According to the comments, the English language of the paper was revised. The modified part is marked in blue in the new upload file.

Reviewer 2 Report

Comments and Suggestions for Authors

I thank the Authors for adequately addressing the majority of my comments. However, I am still not satisfied with the conclusion of Section 3 ("Design and Simulation"). I encourage the Authors to provide a textual explanation of eq. 5 and, more important, a discussion of how their work fits in the state of the art.

Author Response

First of all, thanks for the review. To reflect opinions, an explanation of Equation 5 has been added. And we also added arguments that may be advantageous compared to previous research. Thank you again for your comments on the high quality of the paper.

the English language of the paper was revised according to the opinion of Reviewer 1. The modified part is marked in blue in the new upload file.
